# MD-DFT Calculations on Dissociative Absorption Configurations of FOX-7 on (001)- and (101)-Oriented Crystalline Parylene Protective Membranes

**DOI:** 10.3390/polym16030438

**Published:** 2024-02-05

**Authors:** Weihui Luo, Liang Bian, Faqin Dong, Jianan Nie, Jingjie Yang

**Affiliations:** 1Key Laboratory of Solid Waste Treatment and Resource Recycle, Ministry of Education, South West University of Science and Technology, Mianyang 621010, China; 2Institute of Gem and Material Technology, Hebei GEO University, Shijiazhuang 050000, China; 3State Key Laboratory of Environment-friendly Energy Materials, South West University of Science and Technology, Mianyang 621010, China; 4Tianfu Institute of Research and Innovation, Southwest University of Science and Technology, Chengdu 610299, China

**Keywords:** poly-para-xylylene, membrane, 1,1-diamino-2,2-dinitroethylene, density functional theory, molecular dynamics

## Abstract

Crystalline poly-para-xylylene (parylene) has the potential for use as a protective membrane to delay the nucleation of explosives by separating the explosives and their decomposition products to decrease the explosive sensitivity. Here, molecular dynamics (MD) and density functional theory (DFT) techniques were used to calculate the dissociative adsorption configurations of 1,1-diamino-2,2-dinitroethylene (FOX-7) on (001)- and (101)-oriented crystalline parylene membranes. Based on the results of the calculations, this work demonstrates that the -NO_2_–π electrostatic interactions are the dominant passivation mechanism of FOX-7 on these oriented surfaces. FOX-7 can dissociatively adsorb on oriented parylene membranes due to the interactions between the LUMO of the toluene (or methyl) groups on parylene and the HOMO of the -NO_2_ (or -NH_2_) groups on FOX-7. The formation of a new intermolecular H-bond with the ONO group leads to FOX-7 decomposition via intramolecular C-NO_2_ bond fission and nitro-to-nitrite rearrangement. The most likely adsorption configurations are described in terms of the decomposition products, surface active groups of parylene, binding behaviors, and N charge transfer. Importantly, the (001)-oriented parylene AF8 membrane is promising for use as a protective membrane to passivate the high-energy -NO_2_ bonds during the dissociative adsorption of FOX-7. This study offers a new perspective on the development of protective membranes for explosives.

## 1. Introduction

The novel high-energy material 1,1-diamino-2,2-dinitroethylene (FOX-7, DADNE) was first synthesized in 1998 and instantly received much attention due to its high thermal stability and low impact and friction sensitivities [1]. FOX-7 has a highly polarized carbon–carbon double bond, and the positive and negative charges on the carbons are stabilized by two amino and two nitro groups [2]. It is a nitroenamine with a “push-pull” alkene structure [3]. Under long-term storage conditions, these structures decompose due to the significant aging effects of temperature [4], pressure [5], and pH [6]. NVT (Canonical Ensemble) and AIMD (Ab Initio Molecular Dynamics) simulations revealed that the thermal decomposition mechanism of FOX-7 involves three different initial decomposition steps, namely, C-NO_2_ bond fission and intermolecular and intramolecular hydrogen transfer [7,8]. The C-NO_2_ bond fission mechanism was determined to be the most common route [9]. Because this fission mechanism causes performance issues, particularly for FOX-7 gas and its decomposition products, this explosive requires the use of a protective membrane overcoat that inhibits the slow decomposition [10]. For example, the oxygen atoms of the -NO_2_ groups in FOX-7 partially or completely dissociate from the molecule to oxidize the matrix surface [11]. The FOX-7 molecule decomposes into -ONO and -OC(NO_2_)=CN_2_H_4_, which have unsaturated bonds with high electrostatic potentials that can attract or repel the surface electrons of protective membranes [12]. Hence, a dissociative adsorption mechanism is possible, depending on the particular surface sites involved [13]. Therefore, the protective membranes required for this compound should have low sensitivity, good flexibility, and mechanical strength, and should be incompatible with organic molecules [14].

Recently, it was suggested that polymers can be coated on the surfaces of energetic materials (such as explosives, propellants, and pyrotechnics) to act as protective membranes [15]. Surface crystalline polymers are well known to decrease explosive sensitivity and infiltration [16]. Moreover, the explosive cannot crystallize on the surface of the protective membrane. These membranes can delay the nucleation of the explosive by inhibiting its volumetric expansion [17] and separating the explosive molecules from their decomposition products, hindering heat conduction [18]. Of the various possible polymer membranes, pinhole-free poly-para-xylylene (parylene) and substituted parylene membranes have high stabilities, high chemical and electrical resistances, and low gas permeability [19,20]. Substituted parylene membranes, such as poly-chloro-para-xylylene (parylene C), poly-tetrafluoro-para-xylylene (parylene F), α,α,α′,α′-poly-tetrafluoro-para-xylylene (parylene AF4), and fluorinated poly-para-xylylene (parylene AF8) membranes, were shown to exhibit improved gas impermeability [21]. The primary roles of these membranes are to reduce gas permeation and prevent interactions with organic compounds [22]. The stabilities of fluoroelastomer-coated cyclotrimethylenetrinitramine (RDX) and cyclotetramethylene tetranitramine (HMX) were previously verified [23]. The polar C-Cl and C-F bonds in parylene cause the formation of polymorphic structures [24], including those with (101) [25] and (001) [26] orientations. The exposed C-Cl and C-F bonds are different on the different surfaces [27], which improves the mitigation effects on the explosive thermolysis pathways [28]. Consequently, in this study, based on previous studies of polymer–explosive systems that examined their interfacial structures, phase transitions, heat conduction barriers, etc., four parylene protective membranes were investigated for use as a barrier for FOX-7 and its decomposition products.

Currently, few experimental data are available to explain the efficiency of energetic compounds coated with parylene membranes. Moreover, the mechanism describing the interactions between FOX-7 (or its decomposition products) and parylene surfaces is unknown. To elucidate the fundamental issues related to the nitro-to-nitrite rearrangement of FOX-7 deposited on (101)- and (001)-oriented parylene membranes, this work presents an atomic-level description of the intermolecular hydrogen bonds (H-bonds) formed between the decomposition products of FOX-7 and parylene surfaces. Specifically, MD and DFT calculations were used to study the corresponding local chemical adsorption processes.

## 2. Computational Details

The chain units of various parylene (parylene C, parylene F, parylene AF4, and parylene AF8) membranes were constructed using molecular mechanics (MM), and optimized by 200 ps NPT and NVT (NPT 100 ps before NVT 100 ps) molecular dynamics (MD) simulations at 298 K and 101 KPa using the COMPASS force field [21]. In this work, gradually reduced size (GRS) methods were employed to optimize the weak interaction potentials, reducing the error in the van der Waals and electrostatic potentials between chain units, and regular arrangements of (101)- and (001)-oriented parylene membranes were constructed. These membranes were optimized and relaxed via 110 ps NPT annealing and 500 ps NPT and NVT MD simulations, and the T and P control method was set to Nose Q ratio with 0.01 timestep and Berend decay constant with 0.1 timestep. Short-range van der Waals interactions were calculated using an atom-based method. The Ewald+Group summation method was routinely used to evaluate the long-range electrostatic interactions in reasonably small models [29].

The surface electrostatic energies of the crystalline parylene membranes were determined by density functional theory (DFT) calculations within the generalized gradient approximation (GGA) using the Perdew–Burke–Ernzerhof (PBE) functional. Initially, several tests were performed to verify the accuracy of the method for bulk FOX-7 and its decomposition products on (101)- and (001)-oriented parylene membranes. Specifically, both the physical atomic distributions and chemical electron transfer processes were investigated. The atomic distances and total energies of the models of FOX-7 and its decomposition products on the parylene surfaces were tested for convergence [30,31]. The atomic configurations were optimized by conjugate gradient minimization of the total energy via 500 ps NPT and NVT MD simulations [32]. To elucidate the interactions between the dissociatively adsorbed FOX-7 molecules and the polymer surfaces, the electron localization functions (ELFs) were optimized [11]. The corresponding kinetic energies were calculated by a spin-polarized GGA-PBE DFT method [33]. Other areas of the model and the relative boundary area were simulated by MM using periodic boundary conditions [34].

## 3. Results and Discussion

### 3.1. Surface Structures of the (001)- and (101)-Oriented Crystalline Parylene Protective Membranes

The main parylene polymers used as protective membranes include parylene C ((C_8_H_7_Cl)_n_), parylene F ((C_8_H_7_F)_n_), parylene AF4 ((C_8_H_4_F_4_)_n_), and parylene AF8 ((C_8_F_8_)_n_) [21,29], which are shown in Figure 1b. This figure shows that the errors in the bond lengths and angles are less than 0.5%, which is in contrast to the literature data [35]. The high-energy areas are the most likely sites for the condensation of adjacent chain units to construct low-energy parylene membranes (Figure 1c). Thus, the surface electrons in the -C-Cl (or -C-F), benzyl (π), or -C-F_2_ (or -C-H_2_) bonds of the parylene chain units tend to overlap. For example, the (101) and (001) orientations of parylene C were constructed by exploiting the coupling interactions between the π–π and -C-Cl-Cl-C- bonds, as shown in Figure 1a. The four optimized crystalline parylene configurations are given in Table 1. The calculated crystalline structures and densities are close to the experimental data with errors of less than 1% [21,35]. The errors in the calculated T_g_ values, which depend on the number and nature of the Cl and F atoms, are 4.4%, 6.5%, 8.7%, and 9.4%, which is in contrast to the literature data [21,35,36,37,38,39,40,41].

The parylene surfaces are represented by slab models of these membranes with a 1 nm vacuum. Specifically, (101)- and (001)-oriented parylene membranes with two layers containing 1920 and 2880 atoms were used to investigate the adsorption of FOX-7 and its decomposition products. Typically, the electrons in the benzene π-bonds are the closest to the surface, as shown in Figure 2. Although the chlorine (Cl) and fluorine (F) atoms pull electrons towards themselves [22], the ortho-hydrogen (H) atoms in benzene have the highest electrostatic densities, as shown by the highest occupied molecular orbital (HOMO) analysis. They are the source of the high electrostatic energies (25–70 eV) of parylene C (or parylene F), as shown in Figure 3. However, replacing the H atoms on the benzyl groups with fluorine (F) atoms increases the total surface electron densities [21]. This substitution results in the transfer of the active electrons to the benzyl groups and the formation of the lowest unoccupied molecular orbital (LUMO). The -CF_2_ bonds in parylene AF4 and parylene AF8 have the highest electrostatic energies (350–385 eV). These surface potential energies correspond to positive charges, which facilitate the adsorption of the negatively charged FOX-7 (surface potential energy: −255–2235 eV). 

### 3.2. Adsorption Configurations of FOX-7 on the (001)- and (101)-Oriented Crystalline Parylene Protective Membranes

After determining the potential adsorption sites on the (101)- and (001)-oriented parylene membranes, the possible adsorption configurations of the FOX-7 molecule on these parylene membranes were investigated, as illustrated in Figure 4. These configurations were obtained by defining the distributions of the O and H atoms. The intermolecular H-bond lengths between the -NH_2_ and -NO_2_ groups in FOX-7 are approximately 0.2 nm [12,42]. The -NH_2_ and -NO_2_ bonds rearrange through interactions with adjacent active functional groups, a process that was tested and verified by experiments and simulations [3]. On the surfaces of the FOX-7 molecules, the electrons are transferred between the O-O and H-O bonds through 2p^4^ → 2p^4^ and 1s^1^ → 2p^4^ orbital processes [10,11]. The O-O bonds have high electron transfer ratios, that is, the O-O p → p orbitals have high electrostatic energies that facilitate electron transfer between the adjacent active groups [12]. Figure 5a indicates that the -NO_2_ groups on FOX-7 first bind to the π-bonds of benzene in (101)-oriented parylene C and F or to the benzyl groups in parylene AF4 and AF8. These electrostatic interactions cause the FOX-7 surface electrons to overlap with the parylene membranes, leading to the formation of new short-range intermolecular H-bonds. The adjacent -NH_2_ groups continue to interact with the π-bonds to provide the missing electrons to the -NO_2_-benzene unit until the O-H bonds break [13]. The corresponding energy differences (ΔE) between the adsorption models and single systems are the lowest (parylene C: −11 kcal∙mol^−1^; parylene F: −130 kcal∙mol^−1^; parylene AF4: −7.5 kcal∙mol^−1^; parylene AF8: −10 kcal∙mol^−1^), as shown in Figure 5b. Thus, the FOX-7 molecules lie flat on the (101)-oriented parylene membranes because the HOMO on the -NO_2_ group of FOX-7 interacts with the LUMO on the -C-Cl group on the parylene benzene to lower the energy of the system, as shown by the HOMO–LUMO results presented in Figure 6. The electrostatic interactions are considered to be the dominant adsorption mechanism for FOX-7 on the (101)-oriented parylene membranes. The -NO_2_ bonds (FOX-7), which are the high-energy origin of detonation, are passivated at the two basic adsorption sites, that is, at the toluene and methyl groups, due to the electron transfer from the FOX-7 HOMO to the parylene LUMO [2,3]. Simultaneously, the dissociative adsorption of FOX-7 is also possible and involves the breaking of the intramolecular H-bond in FOX-7 (-NO_2_-H_2_N-) and the formation of a new intermolecular H-bond, which has the smallest calculated bond length (0.2–0.24 nm) with the ONO group.

Another useful tool for investigating the coupling interactions between the FOX-7 active groups and the (101)-oriented parylene surfaces is the electron localization function (ELF). Isosurfaces of the ELF can be calculated and used to explain bond and charge data. The relative energy differences and bond energies are discussed below and are shown in Figure 7. The results of these calculations indicate that both the -NO_2_ and -NH_2_ groups of FOX-7 are bound to toluene groups based on the lowest energy principle. The surface -Cl (or -F) atoms of parylene preferentially capture the surface electrons of the benzene groups, which results in high-energy active electrons that can be transferred to the unsaturated -NH_2_ electron orbitals on FOX-7 [12]. Then, the paired positive holes bind to the negatively charged -NO_2_ groups. In contrast to the electron-withdrawing nature of the -benzene-Cl(F) bonds, the -CF_2_- surface potentials homogenize and push the surface electrons of benzene [23]. These low-energy sites are positively charged and thus preferentially bind to the negatively charged -NO_2_ groups. The surface potentials at the negatively charged methyl bonds are relatively uniform, and these bonds interact with the positively charged -NH_2_ bonds. A common tool for providing a semi-quantitative measure of the charge transfer is the Mulliken charge analysis in which the electronic charge is partitioned among the individual atoms, as shown in Figure 8. Although all the N atoms in the FOX-7 molecule have negative Mulliken charges on the parylene surfaces [43], the N atoms in both the -NH_2_ and -NO_2_ groups have highly negative charges (−2–0.2 e) on the (101)-oriented parylene membranes.

In contrast to the observed interactions between both the -NH_2_ and -NO_2_ groups and the (101)-oriented parylene membranes during adsorption, active methyl groups with positive charges participate in the adsorption process on the (001)-oriented parylene membranes. Based on the lowest energy principle, the exposed paired positive holes on the benzenes can capture the active electrons of the -NO_2_ bonds. The adsorption configurations of FOX-7 on the (001)-oriented parylene AF4 or AF8 membranes are similar to those on the (101)-oriented membranes. However, it should be noted that the -NO_2_ surface electrons are transferred to the exposed low-energy orbitals on the toluene groups on the (001)-oriented parylene C and F membranes, as shown by the lowest energy results presented in Figure 7. Due to the coupling interaction between the high- and low-energy areas of adjacent -benzene-Cl(F) bonds, the exposed methyl groups do not have enough active electrons to break the H-bonds in the FOX-7 molecules. This conclusion is supported by the Mulliken charge results, which show that the positive Mulliken charges are mainly located on the N atoms of the -NO_2_ groups (II2 on parylene C and F: ~0.11 e; II4 on parylene AF4 and AF8: 0.1–0.12 e), as shown in Figure 8. Thus, the four-electron orbitals of the methyl C-H_2_ 2p^2^-1s^1^ (or C-F_2_ 2p^2^-2p^5^) groups mainly capture three surface electrons from the -NO_2_ 2p^3^-2p^4^ orbitals on FOX-7 by forming sp electron orbitals, and the adsorption angles are 45° based on the long-range half-full bonds (complex of seven-electron orbitals and five-electron orbitals (-NH_2_ 2p^3^-1s^1^)), as shown in Figure 5c,d. Because the high-energy -NO_2_ bonds are passivated on the (001)-oriented parylene surfaces, these parylene materials can be used as protective membranes for FOX-7, which is in contrast to the (101)-oriented membranes. Furthermore, the (001)-oriented parylene AF8 is a potential protective membrane for FOX-7 because the binding energies of -NO_2_-benzene and -NO_2_-methyl are the lowest at ~0 eV and 2.5 eV, respectively.

### 3.3. Nitro-to-Nitrite Rearrangement of the Decomposition Products of FOX-7 on the Parylene Membranes

According to Rashkeev et al. [12], FOX-7 molecules decompose via C-NO_2_ bond fission and nitro-to-nitrite rearrangement, and the oxygen atoms in the -NO_2_ groups of FOX-7 partially or completely dissociate to produce (NH_2_)(NO_2_)C=C(NH_2_)O and ONO. Under continuous aging by temperature [4], pressure [5], and pH [6], the -N-O- and -C-O- bonds in the (NH_2_)(NO_2_)C=C(NH_2_)O product rearrange in the same manner as those in FOX-7. The oxygen atoms in the -NO_2_ groups of (a) (NH_2_)(NO_2_)C=C(NH_2_)O partially or completely dissociate to give (b) NH(NH_2_)C=CO and (c) ONO. To analyze the active adsorption sites on the parylene membranes, the probable configurations of the decomposition products of FOX-7 were determined. The probable configurations of the decomposition products on the active adsorption sites (A: methyl (M) and toluene (T) groups) of the parylene membranes are presented as (a1) (NH_2_)(NO_2_)C=C(NH_2_)O and (a2) A-(NH_2_)(NO_2_)C=C(NH_2_)OH in Figure 9; (b1) NH(NH_2_)C=CO, (b2) A-NH(NH_2_)C=CO and (b3) (NH_2_)_2_C=CO in Figure 10; and (c1) HON=O and (c2) O=NO-A in Figure 11. Considering the effects of the active adsorption sites (A: methyl and toluene groups) and orientations of the parylene membranes, the most likely dissociative adsorption processes for the decomposition products of FOX-7 were analyzed based on the lowest binding energy principle. The -NH- and -O=N-O- bonds in the decomposition products have unsaturated orbitals and thus high electrostatic potentials that attract the surface electrons of the exposed toluene and methyl groups of the (101)- and (001)-oriented parylene membranes [11], as illustrated in Figure 12. Thus, active electrons are transferred from the highest occupied molecular orbitals (HOMOs) on the -NH- and -O=N-O- groups of the decomposition products to the lowest unoccupied molecular orbitals (LUMOs) on the active adsorption sites on the parylene surfaces [13]. Therefore, the high-energy -NO_2_ bonds in the decomposition products of FOX-7 are passivated by the toluene and methyl groups, which is consistent with the results for FOX-7.

ELF calculations were used to provide bond information to distinguish between the exposed toluene and methyl groups. The adsorption behaviors involve both -NO_2_–π and -NH_2_–π interactions, which result in (NH_2_)(NO_2_)C=C(NH_2_)O adsorption at the toluene and methyl groups on the (101)- and (001)-oriented parylene surfaces by H-bond (0.15~0.2 nm) fission, as shown in Figure 13a. The secondary dissociative products of (NH_2_)(NO_2_)C=C(NH_2_)O (a), including NH(NH_2_)C=CO (b) and HON=O (c), might also adsorb at the basic toluene and methyl adsorption sites. On the (101)-oriented parylene surfaces, new short-range intermolecular H-bonds (0.21–0.23 nm) involving both the -NH–π and -O=N-O–π (or -NH-methyl and -O=N-O-methyl) hybridized orbitals (b2) form due to electrostatic interactions. This result can be explained by the fact that the unsaturated, hybridized N-H bonds in the -C-NH and -C-NH_2_ groups in (b) have half-full p^4^ (2p^3^-1s^1^) orbitals that can interact with the half-full p^4^ (2p^2^-1s^1^_2_ or 2p^2^-2p^5^_2_) orbitals of the hybridized -C-H_2_ or -C-F_2_ bonds, as shown in Figure 13b. Simultaneously, both the -Cl (or -F) p^7^ orbitals on parylene and the complex p^7^ orbitals (-N-O- 2p^3^-2p^4^ orbitals of the -ONO product) compete to couple with the surface benzene π electrons on the (101)-oriented parylene C and F surfaces, as shown in Figure 13c. The p^7^ (-N-O- 2p^3^-2p^4^) orbital captures an active electron from the hybridized benzene π orbital via -C-F_2_ (or the perfluorobenzene ring) in parylene AF4 (or AF8), creating a hole to offset the electron loss from the -C-F_2_ bond (or perfluorobenzene ring). The corresponding -NO–π and -NH–π bond energies (3.46 and 1.42 e, respectively) are lower than the methyl-ON- bond energies (5.48 and 1.83 e).

In contrast to the observed interactions with the toluene π groups, the half-full hybridized methyl (-C-F_2_) p^4^ (2p^2^-2p^5^_2_) orbitals on the (001)-oriented membranes mainly interact with the half-full p^4^ (N-H 2p^3^-1s^1^) and approximately half-full p^3^ (O-N=O 2p^4^-2p^3^-2p^4^) orbitals on the NH(NH_2_)C=CO and ONO products, respectively. These results are supported by the low energy results for both the -NO–π and -NH–π interactions (V), as shown in Figure 13. The half-full p^4^ and approximately half-full p^3^ orbitals of the -NH_2_ and -NO_2_ bonds, respectively, bind to the active electron–hole sites on the toluene π groups. It should be noted that the positive Mulliken charges (0.09–0.145 e) on the N atoms become negative ((NH_2_)(NO_2_)C=C(NH_2_)O: −1.31–0.25 e; NH(NH_2_)C=CO: −0.34–0.04 e; ONO: −0.36–0.02 e), as shown in the Appendix A. The N 2p^3^ electrons are transferred from the positively charged -NO 2p^3^-2p^4^ orbitals to the negatively charged -NH 2p^3^-1s^1^ (or neighboring O 2p^4^) orbitals through the electronic transitions of the intramolecular -N-C-N- (or -O-N=O) bonds. Consequently, the enhanced -NH orbital strength induced by the (001)-oriented parylene surfaces can delay the nucleation of the two neighboring decomposition products of FOX-7 and inhibit the nitro-to-nitrite rearrangement by hindering intermolecular and intramolecular hydrogen transfer [7]. Therefore, the (001)-oriented parylene AF8 membrane, which has the lowest bond energy, is proposed to be a potential protective membrane for inhibiting FOX-7 decomposition pathways.

## 4. Summary and Conclusions

In conclusion, the dissociative adsorption of FOX-7 on (101)- and (001)-oriented crystalline parylene C, F, AF4, and AF8 membranes was demonstrated. Based on the data provided by this study, it can be concluded that FOX-7 adsorbs on the oriented parylene membranes via -NO_2_–π electrostatic interactions, which appear to be the dominant passivation mechanism of FOX-7. FOX-7 molecules lie flat on the (101)-oriented parylene membranes due to both -NO_2_–π and -NH_2_–π electrostatic interactions, and the -NO_2_ groups mainly bind to the exposed toluene (or methyl) groups on the (001)-oriented parylene membranes. FOX-7 dissociates into (NH_2_)(NO_2_)C=C(NH_2_)O, NH(NH_2_)C=CO, and ONO via C-NO_2_ bond fission and nitro-to-nitrite rearrangement. The high-energy -NO_2_ bonds of (NH_2_)(NO_2_)C=C(NH_2_)O and ONO are passivated by the exposed toluene and methyl groups. In particular, the positively charged -NO 2p^3^-2p^4^ orbitals become negatively charged -NH 2p^3^-1s^1^ (or neighboring O 2p^4^) orbitals through intramolecular -N-C-N- (or -O-N=O) electronic transitions on the (001)-oriented membranes. Based on the ELF and Mulliken charge analysis results, the (001)-oriented parylene AF8 membrane is proposed to be a potential protective membrane for passivating the high-energy -NO_2_ bonds in FOX-7 by a dissociative adsorption process.

The results of this work are insufficient for analyzing the effects of oriented parylene membranes on FOX-7 nucleation and thermolysis mitigation. Further investigations will be focused on the complexation of two FOX-7 molecules and their decomposition products under long-term storage or extreme conditions, such as widely fluctuating temperatures or high pressures. However, this study provides useful information for determining the dissociative processes of FOX-7 on crystalline parylene surfaces and understanding the interactions between explosive products and protective membranes.

## Figures and Tables

**Figure 1 polymers-16-00438-f001:**
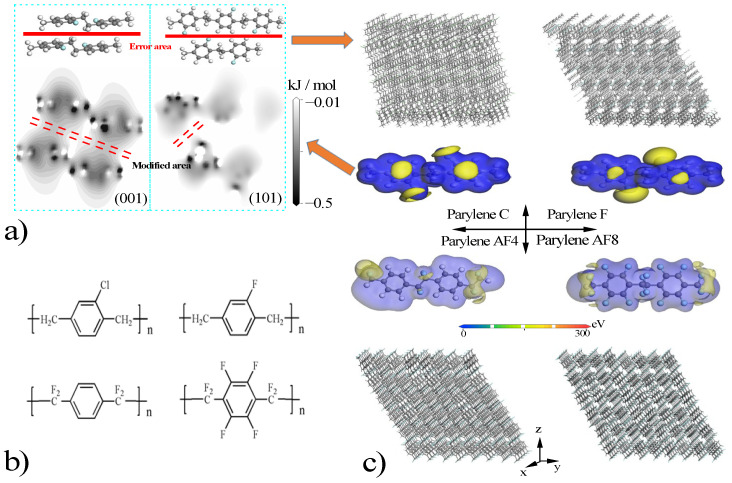
Chain polymerization configurations (**a**), chemical formulas (**b**), potential energy and crystalline models (**c**) of parylene membranes. The color code is the following: grey, C; green, Cl; blue, F and white H. Blue area of parylene chains, low energy area; and yellow area, high energy area.

**Figure 2 polymers-16-00438-f002:**
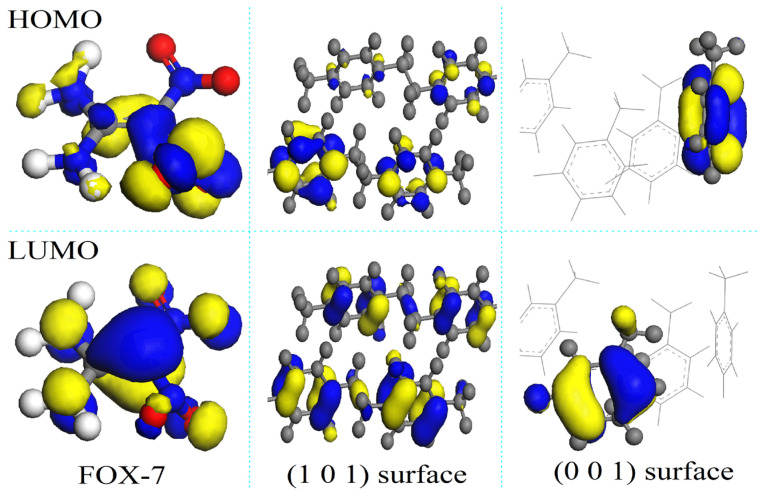
Illustrations of frontier orbitals (Highest Occupied Molecular Orbital: HOMO; Lowest Unoccupied Molecular Orbital: LUMO) of FOX-7, (101)- and (001)-oriented parylene F surfaces. The color code is the following: grey, C; green, Cl; blue, F and white H. Blue area of parylene chains, low energy area; and yellow area, high energy area.

**Figure 3 polymers-16-00438-f003:**
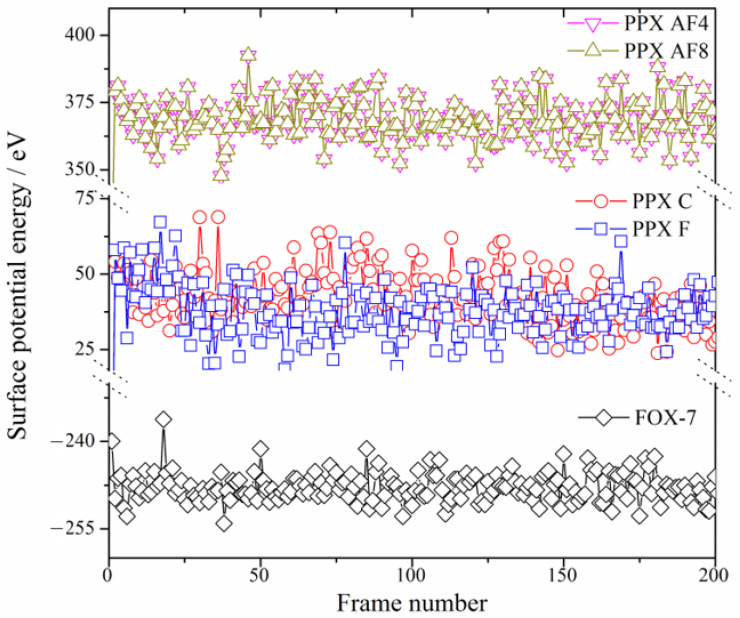
Surface potential energies (eV) of parylene membranes and FOX-7.

**Figure 4 polymers-16-00438-f004:**
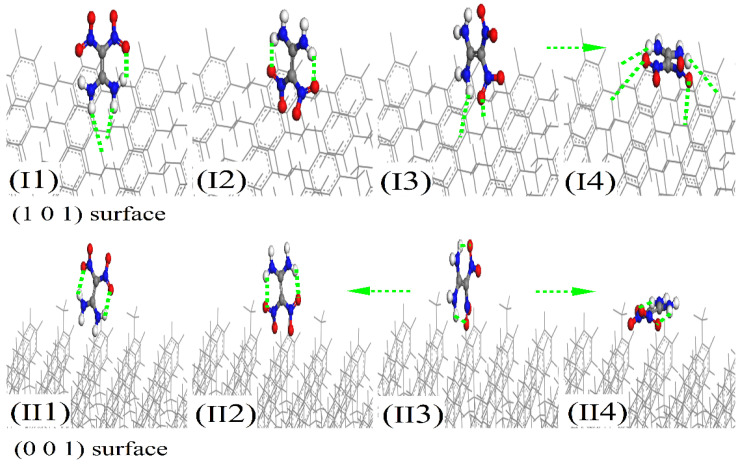
(**I1**–**I4**) Possible adsorption configurations of FOX-7 molecules on the (101)-oriented parylene membranes. (**II1**–**II4**) Possible adsorption configurations of FOX-7 molecules on the (001)-oriented parylene membranes.

**Figure 5 polymers-16-00438-f005:**
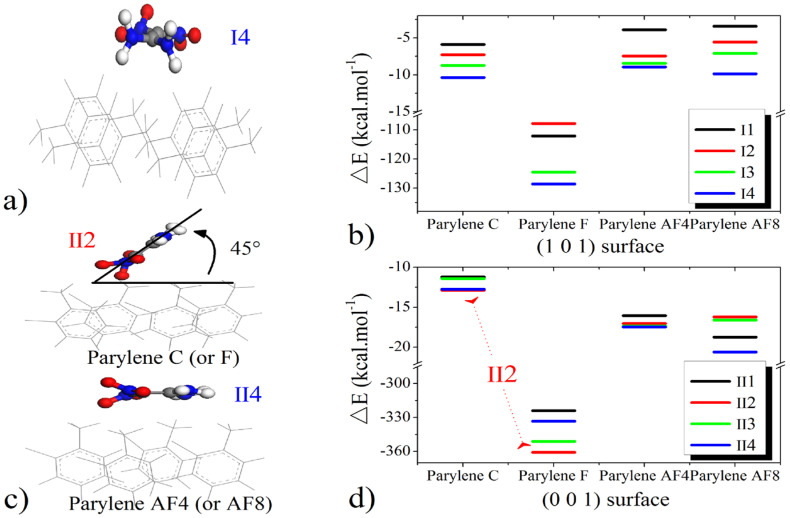
(**a**,**c**) The -NO_2_ groups on FOX-7 bind to the groups in parylene; (**b**,**d**) The corresponding energy differences (ΔE) between the adsorption models and single systems.

**Figure 6 polymers-16-00438-f006:**
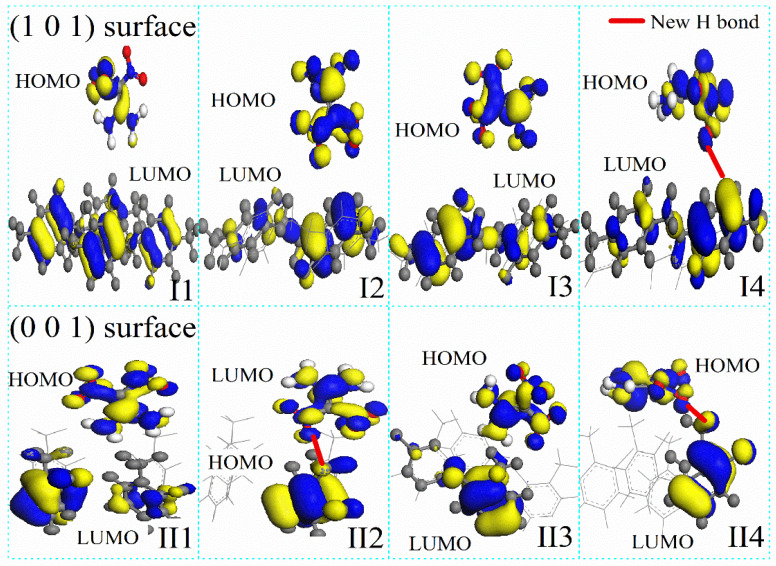
Illustrations of frontier orbitals of FOX-7 on (101)- and (001)-oriented parylene F membranes. (**I1**–**I4**) Possible adsorption configurations of FOX-7 molecules on the (101)-oriented parylene membranes. (**II1**–**II4**) Possible adsorption configurations of FOX-7 molecules on the (001)-oriented parylene membranes.

**Figure 7 polymers-16-00438-f007:**
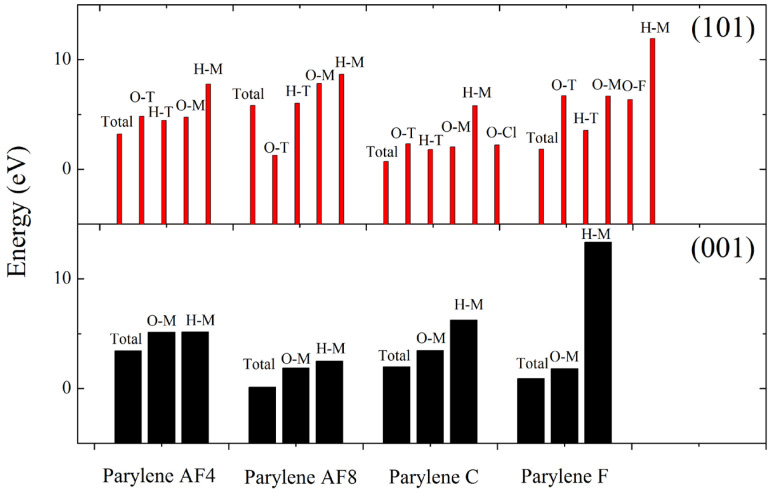
Binding energies (eV) of various probable configurations of FOX-7 adsorbed on (101)- and (001)-oriented parylene membranes. In the figure, “M” refers to the methyl group, “T” refers to the toluene group, and “O” and “H” refer to the -NO_2_ and -NH_2_ functional groups of FOX-7, respectively.

**Figure 8 polymers-16-00438-f008:**
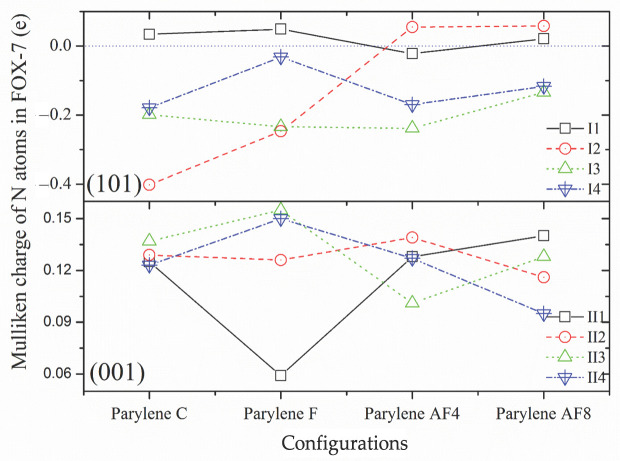
Mulliken charges of total N atoms of FOX-7 on (101)- and (001)-oriented parylene membranes.

**Figure 9 polymers-16-00438-f009:**
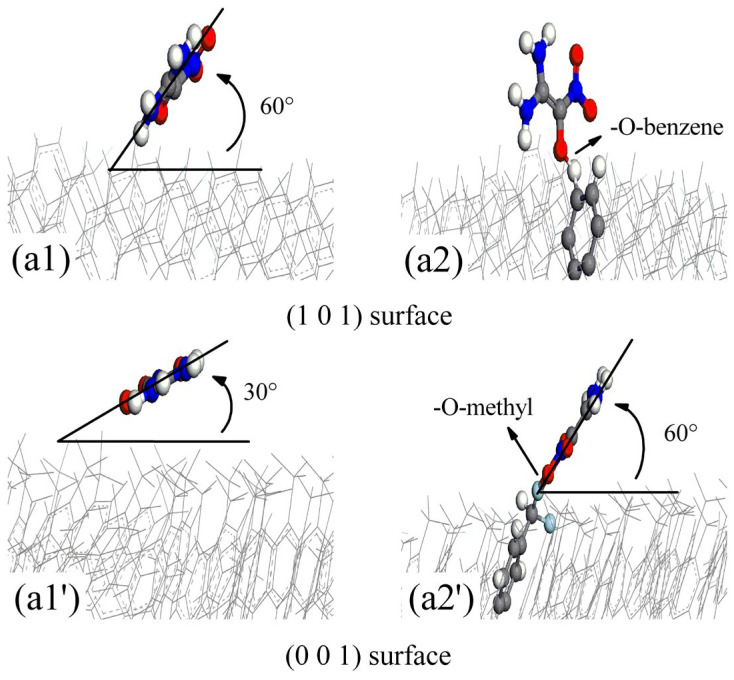
The most probable absorption configurations of (NH_2_)(NO_2_)C=C(NH_2_)O and A-(NH_2_)(NO_2_)C=C(NH_2_)OH on (101)- and (001)-oriented parylene membranes. In the figure, “A” refers to active adsorption sites of parylene surfaces. (**a1**,**a1′**) (NH_2_)(NO_2_)C=C(NH_2_)O; (**a2**,**a2′**) A-(NH_2_)(NO_2_)C=C(NH_2_)OH.

**Figure 10 polymers-16-00438-f010:**
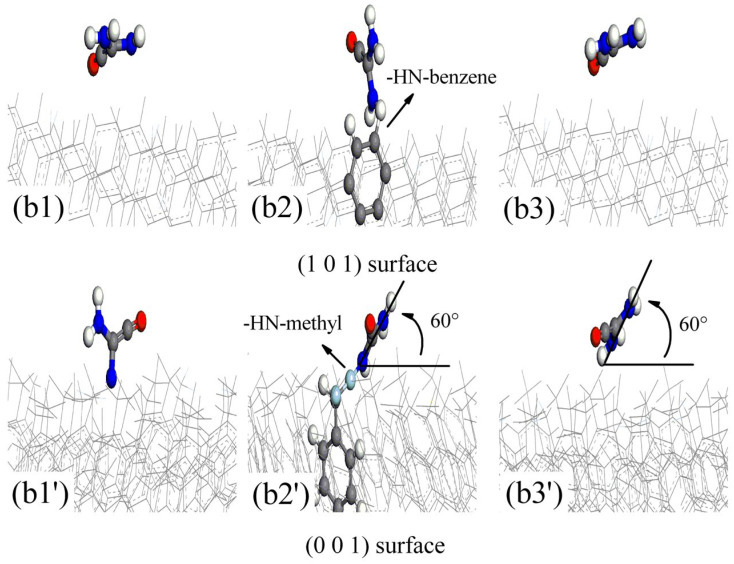
The most probable absorption configurations of NH(NH_2_)C=CO, A-NH(NH_2_)C=CO, and (NH_2_)2C=CO on (101)- and (001)-oriented parylene membranes. (**b1**,**b1′**) NH(NH_2_)C=CO; (**b2**,**b2′**) A-NH(NH_2_)C=CO; (b3, b3′) (NH_2_)2C=CO.

**Figure 11 polymers-16-00438-f011:**
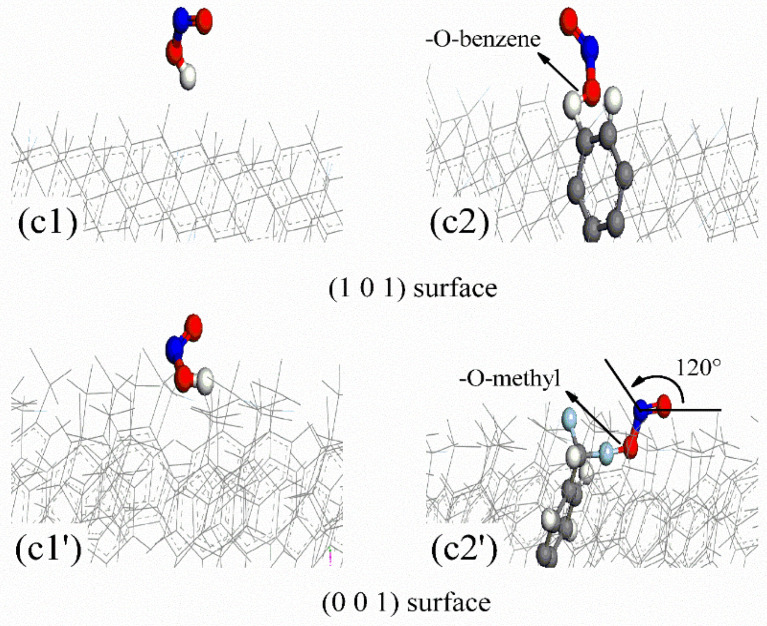
The most probable absorption configurations of HON=O and O=NO-A (FOX-7) on (101)- and (001)-oriented parylene membranes. (**c1**,**c1**′) HON=O; (**c2**,**c2′**) O=NO-A.

**Figure 12 polymers-16-00438-f012:**
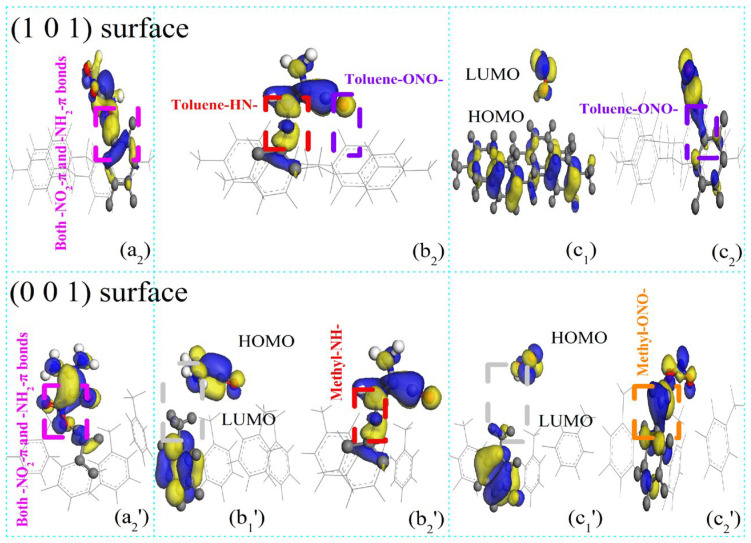
Illustrations of frontier orbitals of FOX-7’s decomposition products on (101)- and (001)-oriented parylene F membranes. (**a_2_**,**a_2_′**) A-(NH_2_)(NO_2_)C=C(NH_2_)OH; (**b_1_′**) NH(NH_2_)C=CO; (**b_2_**,**b_2_′**) A-NH(NH_2_)C=CO; (**c_1_**,**c_1_′**) HON=O; (**c_2_**,**c_2_′**) O=NO-A.

**Figure 13 polymers-16-00438-f013:**
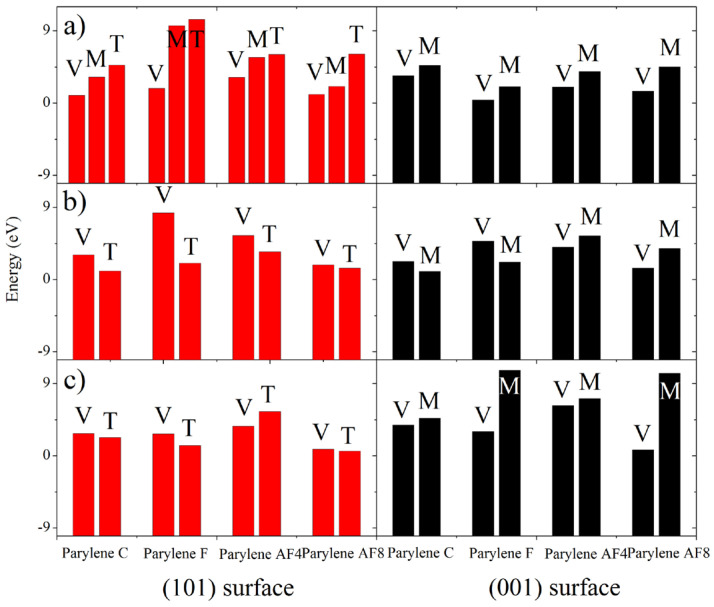
Binding energies (eV) of various probable configurations of FOX-7’s decomposition products adsorbed on (101)- and (001)-oriented parylene membranes. (**a**–**c**) reflect the adsorption models of parylene-(NH_2_)(NO_2_)C=C(NH_2_)OH, parylene-NH(NH_2_)C=CO, and O=NO-parylene, respectively. In the figure, “V” refers to the adsorption behavior of both -NO–π and -NH–π interactions of FOX-7’s decomposition products being bound by parylene toluene groups through intramolecular H-bond breakage, “M” refers to methyl-ON-(or -NH-) adsorption, and “T” refers to toluene-ON-(or -NH-) adsorption.

**Table 1 polymers-16-00438-t001:** Calculated structures, densities, and glass transition temperatures (T_g_) of parylene membranes. “a–c” show the lengths of parylene cells.

	Parylene C [36,37,38]	Parylene F [39,40]	Parylene AF4 [41]	Parylene AF8 [21]
a (nm)	0.95	0.88	1.15	1.17
b (nm)	2.15	2.15	1.95	2.13
c (nm)	4.21	4.64	3.72	3.85
Density (g·cm^−3^)	1.29 (1.27)	1.13 (1.11)	1.71 (1.58)	2.17 (1.93)
T_g_ (K)	332 (347)	312 (289)	840 (773)	610 (673)

The experimental and computer data are tested by SEM and MD results, as shown in Refs. [21,35,36,37,38,39,40,41].

## Data Availability

The data that support the findings of this study are available from the corresponding author, Liang Bian, upon reasonable request.

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
