# Peer review of "MD-DFT Calculations on Dissociative Absorption Configurations of FOX-7 on (001)- and (101)-Oriented Crystalline Parylene Protective Membranes"

_polymers, 2024, doi:10.3390/polym16030438_

Round 1
Reviewer 1 Report
Authors have studied the dissociative adsorption capability of FOX-7 on crystalline parylene membrane to reduce the explosive sensitivity. For this purpose authors have used MD and DFT as the material model. Authors have found that NO_2-\pi electrostatic interaction are the dominant passivation mechanism of FOX-7. The findings of this research would help to design better protective membrane for the explosives. Authors suggests that (001) oriented parylene AF8 membrane might be a promising candidate for use as a protective membrane for FOX-7 explosives.
This research is interesting and will be useful for community who works in the high entropy explosives field.
Following are the suggestions to make the present works and presentations better:
1. In the introduction, authors have mentioned that "Under long-term storage conditions, these structures decompose due to the significant aging effects of temperature,pressure and pH." So it's basically an aging effect. The time scale that can be achieved by MD or DFT is much much smaller. Any comment on that?
2. In line 86, initial parylene structure are optimized using MD with (NPT+NVT) ensemble and COMPASS force filed. What is the target temperature and pressure used?
3. Numerical implementation of NPT and NVT ensemble involved some numerical parameters, e.g, damping parameter for T and P. Provide those values used in this simulation.
4. While using MD to optimize different paryleme structure, did you use (NPT+NVT) together or successively. If used together then all (P,V,T) are constrained, how energy will get minimized?
5. Figure-1 seems to have bad aspect ratio and low resolution. Quality of the figure is not of publication standard.
6. In line 128, "as shown in “()”. reference is missing.
Author Response
Comment 1: In the introduction, authors have mentioned that "Under long-term storage conditions, these structures decompose due to the significant aging effects of temperature,pressure and pH." So it's basically an aging effect. The time scale that can be achieved by MD or DFT is much much smaller. Any comment on that?
Response 1:The focus of this article is on the interaction between FOX-7 decomposition products and PPX, which are the products after aging and decomposition. The aging process is not the main concern of this article. In addition, MD calculations describe the reaction process and do not correspond to the experimental time scale, but they can reflect the regularity of the process.
Comment 2: In line 86, initial parylene structure are optimized using MD with (NPT+NVT) ensemble and COMPASS force filed. What is the target temperature and pressure used?
Response 2:The temperature used is 298K and 101kPa, already added to the text and highlighted in red (see line 92).
Comment 3: Numerical implementation of NPT and NVT ensemble involved some numerical parameters, e.g, damping parameter for T and P. Provide those values used in this simulation.
Response 3:These parameters already added to the text and highlighted in red (see line 92).
Comment 4: While using MD to optimize different paryleme structure, did you use (NPT+NVT) together or successively. If used together then all (P,V,T) are constrained, how energy will get minimized?
Response 4:NPT first, NVT second, the expression in the text has been modified and and highlighted in red (see line 91)
Comment 5: Figure-1 seems to have bad aspect ratio and low resolution. Quality of the figure is not of publication standard.
Response 5 :The figure has been modified as requested.
Comment 6: In line 128, "as shown in “()”. reference is missing.
Response 6:Thank you for the correction, the sentence has been modified.
Reviewer 2 Report
The proposed research work mainly proposes the interaction of FOX and their dissociated products on the parylene materials. Although, the authors provided several pieces of information still the draft is not clear enough to publish in polymers and needs significant improvement prior to the publication.
Line 36 explains “ four amino and nitro groups” which misleads and therefore it should be “contains two amino and two nitro groups”
Line 38, write expansions for NVT and AIMD
Although the authors cited the work in the introduction section, the name of the systems is missing and they have to mention the name of the materials that were used in the cited works. For instance, Line 53, can be coated on the surfaces of energetic materials (what are those energetic materials). This information can help the reader to look at the work interestingly without scrolling for other cited works.
Line 81, it should be MD and DFT, not MD-DFT
Line 84, how many chain units are constructed with dimensions/size?
Line 86, Did the authors combine both NVT+NPT? How? I suppose these are different methods? More elaborate discussion is required including damping constant and temperature, and pressure.
What about the timestep for MD?
Mention/Show the simulation box with axis and sizes in all directions.
The authors mentioned annealing. Was the temperature varied in the simulation?
In DFT, which software the authors was used?
What about the basis set? Were these DFT calculations performed with pseudopotential code?
Strongly suggest rewriting the computational method section which should contain how authors prepared and run the systems with MD and DFT. From the methods described in the section, it is very hard to adopt and redo the same calculation by someone. Not clear enough.
Figure 1 needs to be enhanced as it is not legible enough. Increase the size and clarity.
How the authors calculate the Tg is totally missing in the method section. The calculation of Tg is rather difficult using MD and there have been several works tried to account for such temperature and still, this is a big debate. Strongly advise providing the method and the fitting curves which can be used for Tg calculation. Otherwise, those values are not correct to be published. What about standard deviations in density and Tg? How many different simulations are being used as a starting point?
Figure 4 caption is not correct
Figure 4 should be enhanced with color and bond size should be increased
Figure 5 caption is again wrong
Too many figures in the manuscript and some figures can be placed in the supporting information.
Mulliken charges are hardly used to make conclusions as it fluctuates highly with respect to basis sets and methods used in DFT. Instead, other charges can be proposed, including chelpg, resp or bader etc.
Overall, the manuscript should be majorly improved to make it more interesting for the scientific community. Furthermore, almost all figures should be improved and the captions and texts must be checked by all authors prior to the submission.
Author Response
Comment 1: Line 36 explains “ four amino and nitro groups” which misleads and therefore it should be “contains two amino and two nitro groups”
Response 1:Thank you for the correction, the sentence has been modified and highlighted in red (see line 38).
Comment 2:Line 38, write expansions for NVT and AIMD.
Response 2:Corresponding information has been added to the article,and highlighted in red (see line 40).
Comment 3:Although the authors cited the work in the introduction section, the name of the systems is missing and they have to mention the name of the materials that were used in the cited works. For instance, Line 53, can be coated on the surfaces of energetic materials (what are those energetic materials). This information can help the reader to look at the work interestingly without scrolling for other cited works.
Response 3: Thank you for your suggestion.,the information has been added to the article,and highlighted in red (see line 57).
Comment 4:Line 81, it should be MD and DFT, not MD-DFT
Response 4: Thank you for the correction, the sentence has been modified,and highlighted in red (see line 84 and 97).
Comment 5:Line 84, how many chain units are constructed with dimensions/size?
Response 5:The dmensional information of the polymers has been supplemented in Table 1,and highlighted in red.
Comment 6:Line 86, Did the authors combine both NVT+NPT? How? I suppose these are different methods? More elaborate discussion is required including damping constant and temperature, and pressure.What about the timestep for MD?Mention/Show the simulation box with axis and sizes in all directions.The authors mentioned annealing. Was the temperature varied in the simulation?In DFT, which software the authors was used?What about the basis set? Were these DFT calculations performed with pseudopotential code?Strongly suggest rewriting the computational method section which should contain how authors prepared and run the systems with MD and DFT. From the methods described in the section, it is very hard to adopt and redo the same calculation by someone. Not clear enough.
Response 6:Thanks for your suggestion, In DFT, MS softwera CASTEP module was used, and some details have been added to the calculation method section, including information on temperature, pressure, and damping parameters and step sizes, and the less rigorous descriptions such as "NVT+NPT" have been modified.
Comment 7:Figure 1 needs to be enhanced as it is not legible enough. Increase the size and clarity.
Response 7:Thanks for pointing this out, all figure have been modified.
Comment 8:How the authors calculate the Tg is totally missing in the method section. The calculation of Tg is rather difficult using MD and there have been several works tried to account for such temperature and still, this is a big debate. Strongly advise providing the method and the fitting curves which can be used for Tg calculation. Otherwise, those values are not correct to be published. What about standard deviations in density and Tg? How many different simulations are being used as a starting point?
Response 8:Thank you for your valuable comments. In this article, Tg is calculated by finding the inflection point of the rate of volume change with increasing temperature, and the correctness of the Tg values is verified by experimental data. The focus of this part of the article is on the error rates obtained by comparing the model (ideal model) with the actual molecular structure (with defects) based on the actual density modeling and after several structural relaxations,as shown in “Ref [21, 35, 36~41]”. The method of calculating Tg used in the article only shows that polymers similar to the four polymers used in this article may be able to obtain Tg values close to the actual situation by this method.
Comment 9:Figure 4 caption is not correct. Figure 4 should be enhanced with color and bond size should be increased. Figure 5 caption is again wrong. Too many figures in the manuscript and some figures can be placed in the supporting information.
Response 9:Thanks for the suggestion, it's very useful and Figures 4 and 5 have been modified accordingly and the overall number of images has been scaled down.
Comment 10:Mulliken charges are hardly used to make conclusions as it fluctuates highly with respect to basis sets and methods used in DFT. Instead, other charges can be proposed, including chelpg, resp or bader etc.
Response 10:The Mulliken charge data can illustrate a portion of the charge transfer pattern by comparison. Thank you for the suggestion, and in order not to affect the overall structure of the article, we have placed this data in the supporting information.
Reviewer 3 Report
The Manuscript "MD-DFT calculations on dissociative absorption configurations of FOX-7 on (001) and (101) orientated crystalline parylene protective membranes" can be published after some minor corrections.
Quality of the Figures has to be improved, practically all of them are low resolution and sometimes not readable enough. Recheck for typos in the text.
In general, the manuscript deals with a well performed computer simulation study, it is easy to follow and understand. The background, motivation, aims and simulation details are well explained. The Authors are realistic about the conclusions and provide a context for future work. Therefore I would recommend this paper to be published, assuming that the technical remark regarding figures has been addressed.
Author Response
Comment: Quality of the Figures has to be improved, practically all of them are low resolution and sometimes not readable enough. Recheck for typos in the text.
Response:Thank you for your comments, have revised the images and checked the manuscript as requested.
Round 2
Reviewer 2 Report
I appreciate the authors for taking into account all comments.